# Light-Based IoT: Developing a Full-Duplex Energy Autonomous IoT Node Using Printed Electronics Technology

**DOI:** 10.3390/s21238024

**Published:** 2021-12-01

**Authors:** Malalgodage Amila Nilantha Perera, Marcos Katz, Juha Häkkinen, Roshan Godaliyadda

**Affiliations:** 1Centre for Wireless Communications, University of Oulu, 90570 Oulu, Finland; marcos.katz@oulu.fi; 2Circuits and Systems Research Unit, University of Oulu, 90570 Oulu, Finland; juha.hakkinen@oulu.fi; 3Department of Electrical and Electronic Engineering, University of Peradeniya, Kandy 20400, Sri Lanka; roshangodd@ee.pdn.ac.lk

**Keywords:** LIoT, VLC, energy-autonomous, printed electronics, zero-energy, IoT

## Abstract

The light-based Internet of things (LIoT) concept defines nodes that exploit light to (a) power up their operation by harvesting light energy and (b) provide full-duplex wireless connectivity. In this paper, we explore the LIoT concept by designing, implementing, and evaluating the communication and energy harvesting performance of a LIoT node. The use of components based on printed electronics (PE) technology is adopted in the implementation, supporting the vision of future fully printed LIoT nodes. In fact, we envision that as PE technology develops, energy-autonomous LIoT nodes will be entirely printed, resulting in cost-efficient, flexible and highly sustainable connectivity solutions that can be attached to the surface of virtually any object. However, the use of PE technology poses additional challenges to the task, as the performance of these components is typically considerably poorer than that of conventional components. In the study, printed photovoltaic cells, printed OLEDs (organic light-emitting diodes) as well as printed displays are used in the node implementation. The dual-mode operation of the proposed LIoT node is demonstrated, and its communication performance in downlink and uplink directions is evaluated. In addition, the energy harvesting system’s behaviour is studied and evaluated under different illumination scenarios and based on the results, a novel self-operating limitation aware algorithm for LIoT nodes is proposed.

## 1. Introduction

The Internet of things (IoT) paradigm aims at providing wireless connectivity to virtually any object, such as machines, specialised sensors and actuators, home and office appliances, and everyday items. The number of connected IoT devices is rapidly growing, and it is estimated a several-fold increase in the number of these devices by the end of this decade from the current 10 billion (2021). Providing wireless connectivity to a massive number of IoT nodes needs to be performed in an efficient manner, particularly considering the use of key resources such as energy, spectrum and network infrastructure. Moreover, the environmental impact of such a massive connectivity should be minimised. Thus, IoT technology should be designed not only to be an effective communication system but also a sustainable one. Clearly, the design of both network infrastructure and IoT nodes needs to be carefully considered to fulfil these goals. 

To this end, we consider in this paper the concept of light-based IoT (LIoT) as a sustainable IoT approach to provide wireless connectivity. LIoT exploits the zero-energy node concept, where an energy-autonomous node harvests from the environment the energy needed to operate. Light is exploited in the LIoT concept not only to power up the node but also to transfer information to and from the IoT node. Similar approaches have been studied in [1]. In general, all these approaches exploit the lighting infrastructure to also provide wireless connectivity, and they have the well-known advantages of optical wireless communications, such as the available extensive unlicensed spectrum, inherent security, and interference-free operation. A crucial aspect of the LIoT concept studied here is that its implementation in a sustainable manner is investigated. For that purpose, printed electronics (PE) technology is used in the implementation of the concept. In the future, we envision that fully printed IoT nodes will be implemented on a thin substrate (e.g., paper, plastic. etc.), resulting in a very low cost, highly flexible and sustainable connectivity solution that could be attached to any object. As such, LIoT is promising not only for conventional IoT applications but also for novel use cases, including logistics, industrial, healthcare and others. The prospect of producing a small form factor layer containing a complete self-powered IoT node with some functionalities is truly appealing, and in this paper, we discuss the design, implementation and performance evaluation of such a LIoT node using PE technology. Given the current state of PE technology, it is not possible today to implement a fully printed LIoT node, a task which remains our long-term ultimate goal. However, in this research, we make partial use of PE, employing several critical components using printed technology, including photovoltaic (PV) cells, light-emitting diodes (LEDs) and displays. The fact that the performance of PE components is typically poorer than that of conventional electronic components is one of the key challenges of LIoT. Particularly, since energy harvesting is vital for LIoT, the fact that the efficiency of printed PV cells is considerably smaller than that of typical cells requires careful system design, as it will be discussed in this paper. Moreover, it is shown that efficient energy harvesting, storing and managing are essential tasks that need to be mastered in order to successfully make LIoT a reality. The paper discusses the detailed design and implementation of a full-duplex LIoT node using PE technology, taking into account the design constraints posed by that technology. 

The main contributions of this work can be summarised as follows. Initially, this paper introduces and discusses the unique advantages, the applicability of the Light-based IoT concept, and the potential of implementing LIoT using printed electronics. Moreover, the paper proposes using methods and essential technologies such as DC-DC converters, energy harvesting protocols, MPPT and modulations to improve the dual-mode operation (energy harvesting and communication) of LIoT utilising indoor illumination. By selecting suitable technologies and methods from the above mentioned, a LIoT based smart label prototype was implemented as a case study. Based on the prototype’s performance, the practicality of the proposed framework, effect of using directed and non-directed radiation patterned optical links were analysed. In addition, a novel self-operating limitation aware algorithm for LIoT was proposed allowing the node to decide its operating mode (e.g., active/sleeping) based on the balance between its current energy capabilities and the task energy requirements.

There are numerous research studies that have been carried out to create energy-autonomous IoTs using renewable energy sources. According to the survey conducted in [2], energy sources such as ambient light, wind, vibration, thermal, motion energy and radiofrequency can be used for IoT energy harvesting purposes. In addition, according to the study, ambient light can be considered as the most efficient renewable source with the highest efficiency of consistent energy conversion capabilities compared to other sources.

Therefore, there is an extensive range of possible applications for ambient light energy harvesting based IoT designs. In [3], authors propose energy-autonomous radio frequency (RF) based IoT proof of concept with a solar energy harvesting approach. The energy harvesting system proposed in the design consists of traditional Si rigid photovoltaics and lithium-ion batteries capable of catering to the power requirement of the Wi-Fi, sensors, and microcontroller units. The node is designed to harvest energy during the daytime and save the required energy in the battery. In order to keep the node in a low power consuming state during idle periods, the use of duty cycle schedule-based sleep and wakeup mechanism was exploited in the work. Moreover, the authors propose a microcontroller-based adaptive duty cycling schedule that changes according to the system’s solar irradiance and battery levels. In [4], the authors propose an energy-autonomous IoT design using the visible light backscatter approach. The node is initially powered up by photovoltaic energy harvesting and expected to communicate using visible light during the operation. The node’s uplink is activated by an electronically activated liquid crystal display (LCD) shutter which is designed to expose the retroreflector according to the uplink data modulation. The IoT node is operating passively; hence, no energy is stored during any part of the operation.

The designs from the above-discussed works limit the practicality of integrating these with many modern applications. Using traditional electronics and high energy density, chemical reaction-based batteries for implementation could introduce limitations on IoT applications requiring compact form factor and chemical hazardous-free designs. Even though the nodes are expected to operate maintenance-free for a long lifetime, the limited number of battery cycles can constrain the product’s life span at some point. Approaches such as retroreflectors and RF antenna-based communication can be less advantageous when the designs are required to fulfil modern internode communication and power-sharing without adding additional hardware/circuitries. In addition, a study in [5] shows that long term Wi-Fi radiation exposure can cause minor health risks even if they are within the standard guideline exposure range. Therefore, it is essential to address the modern IoT requirements and features such as internode data /power-sharing, small compact designs with flexibility characteristics, low manufacturing costs, minimum electromagnetic radiation exposure to users and environmental impact at the design stage of the IoT technology.

In the presence of these considerations, this paper is organised as follows. Section 2 introduces in detail the LIoT concept and discusses some interesting applications. In Section 3, a brief overview of key technologies for sustainable implementation are introduced. Section 4 discusses requirements and identifies research challenges. The design of a LIoT node supporting uplink and downlink communications is described in Section 5, while in Section 6 performance of the implemented LIoT node is assessed. Finally, Section 7 concludes the paper.

## 2. LIoT Concept

The Light-based IoT concept proposes a novel design for IoT which enables light to be used to energise and to provide wireless connectivity to an IoT node. This concept follows the nature of “expose and connect”, simply describing the operating principle: when a node is exposed to light, it will be connected to the Internet [1]. According to the LIoT concept, indoor photovoltaic energy harvesting technology is utilised for the node’s power requirement, making the nodes entirely energy-autonomous. At the same time, VLC is used to wirelessly connect with the network. As an advantage of this technology, the indoor lighting infrastructure used for indoor illumination can be simultaneously used for IoT networking purposes. In addition, inter-node communication, localisation awareness and power-sharing technologies such as optical wireless power transfer (OWPT) can be possibly integrated with the concept-based nodes, making them comparative to the traditional IoTs. The structure of LIoT node is depicted in Figure 1.

The dual-purpose transmitter is expected to transmit modulated data to the LIoT nodes creating a visible light communication link while providing flicker-free eye-friendly indoor illumination. The LIoT node is designed to receive the optical signal through the optical channel while harvesting the energy from the incident irradiation on the integrated PV cells. Depending on the optical signal detecting performance, a sensitive photo sensor or energy harvesting integrated PV cells can perform the receiving task.

Similarly to the operation of the transmitter, the visible light transmitter integrated on the node can be used for both power-sharing and communication with neighbouring nodes. In that way, this technology can be used to communicate and power up wireless sensor networks simultaneously and cooperatively. In addition, integration of location awareness on the node based on the received optical signal could further improve inter-node communication and power-sharing efficiency. Awareness of the proximity of neighboring nodes can be used to improve internode communication and power transferring capability of the wireless sensor network. 

There are several unique benefits that can be attained from applying this technology to future IoT. Since VLC is used for communication, all unique advantages of VLC, such as license-free spectrum with unlimited reusability, robustness to electromagnetic interference and inherent physical security, are valid for the LIoT nodes. In addition, with the battery-free energy-autonomous design, the nodes implemented with LIoT technology are expected to operate without any regular maintenance. These promising characteristics will pave the way towards future green IoT communications. Moreover, the LIoT concept lends itself to the development of sustainable IoT solutions, particularly when implemented with novel technologies such as printed electronics and others, as will be discussed in the paper.

### Applications

LIoT can be applied in most of the conventional IoT scenarios. In addition, there are some applications that particularly match LIoT characteristics. A few compelling applications of LIoT technology are briefly discussed next to highlight its potential. LIoT can improve the quality of the logistics sector. With the use of sustainable implementation technology such as printed or biodegradable electronics, very low-cost smart labels or container-based IoT nodes can be made to monitor and sense the environment and to display relevant real-time information. Since LIoT nodes are battery-free, the risk of fire hazards and quality degradation over time is minimal. The transceivers can be directly integrated with the warehouse/storage areas existing lighting infrastructure without costly modifications. Easily integrable sensors such as temperature, humidity, and colour can be used to monitor each product condition in real-time while supporting novel adaptive storage climate controlling systems. The low power flexible displays can be used to display real-time updating information such as price, discounts, condition-based expiry dates or even on product advertisement platforms. The self-localisation and proximity awareness of the nodes can be utilised to improve the operations of management and supply services. In addition, LIoT nodes can operate harmlessly even in electromagnetic sensitive areas such as hospitals, aviation, or manufacturing plants. Low-cost implementation (with existing lighting infrastructure), long time reusability, electromagnetic interference- and exposure-free robust operation, ability of supporting large range of sensor applications and maintenance-free uninterrupted operation can be highlighted as some of the key advantages that can be obtained from integrating LIoT technology.

The duplex LIoT design can be used to replace the traditional design of the day-to-day devices such as headphones, hearing aids and museum guides. Since VLC has the ability to reach higher data rates, traditional radio frequency technologies used for portable devices such as Bluetooth can be replaced with LIoT technology. In this way, users can use these products wherever LIoT friendly indoor illumination is available.

The underwater VLC (UVLC) is a promising technology which has already proven to go beyond acoustic signal based underwater communication [6,7]. With UVLC friendly modulations, more robust communications links can be made with underwater applications [8]. Since recharging or replacing the batteries of battery-based IoTs is a tedious and challenging process, moving to the energy harvesting design approach is considered a more feasible solution in the long run. In [9] authors propose a hybrid optical and acoustic communication technique with an energy harvesting approach. Similarly, by combining UVLC technology with energy-autonomous LIoT, underwater IoT quality of service can be further improved. In contrast, the LIoT designs’ less energy-autonomous nature becomes more advantageous than traditional designs when it comes to underwater IoT operations.

## 3. Printing Electronics for a Lower Environmental Impact

Recent years have seen a lot of interest in printed electronics, a type of electronics created using conventional printing techniques. The key advantages of PE include low-cost manufacturing with high throughput, compatibility with system fabrication on flexible or stretchable substrates utilising inorganic, organic, and bioinspired materials, and relative ease of integration of complete systems or hybrid systems on a single sheet of plastics or even paper. Contrary to this, silicon-based microfabrication processes such as photolithography and vacuum-based methods (e.g., sputtering, chemical vapour deposition) tend to be complex and costly. A variety of applications have been enabled by printing functional inks containing soluble or dispersed materials on PE, such as transparent conductive films, flexible energy harvesting, thin-film transistors, electroluminescent devices, and wearable sensors [10].

### 3.1. Manufacturing Techniques

Printing electronics and sensors traditionally involves bringing pre-patterned parts of a circuit in contact with flexible substrates and transferring functional inks or solutions onto them. The two major approaches for developing printing systems are contact and non-contact printing. Contact printing involves bringing patterned structures with inked surfaces into physical contact with the substrate. Non-contact processes deploy the solution through openings or nozzles arranged in a pre-programmed pattern, while structures are defined based on the movement of the stage (substrate holder). Contact-based printing technologies include gravure printing, gravure-offset printing, flexographic printing, and R2R printing. Inkjet printing, screen-printing, and slot-die coating are among the non-contact printing methods. In the non-contact printing field, the advantages are several. These include simplicity, affordability, speed, adaptability to fabrication processes, low material waste, high precision of patterns, and easy control by adjusting just a few process parameters. The newly emerging polymeric stamp-based printing methods such as nanoimprint, micro-contact printing, and transfer printing have also generated significant interest, especially for monocrystalline semiconductor-based flexible electronics. 

According to [11], printing technologies such as gravure and slot-die can achieve 50–200 µm high print resolution while printing at 0.02–12 µm and 0.15–60 µm thickness levels respectively. On the other hand, flexographic and screen printing technologies are more capable of achieving beyond 100 m/min printing speeds. Technologies such as Inkjet and transfer printing have the unique advantage of low material wastage compared to other methods.

### 3.2. Materials

In order to build (all) printed systems, both functional inks suitable for different printing techniques and flexible substrates described next are needed.

#### 3.2.1. Functional Inks

Typical applications for conductors include wiring, interconnects, electrodes, antennas, touch buttons, etc. In general, copper, gold, and aluminium are the most common materials used. Silver is the most widely used printed conductor material, and there are many commercial silver inks available for numerous printing processes. Printed electronics use both nanoparticle and microparticle silver inks. Compared to microparticle silver inks, nanoparticle inks have typically better conductivity but are also usually more expensive. The much higher resistivity of both nano and microparticle inks compared to solid metal conductors is perhaps the biggest difference when designing printed vs. printed circuit board-based electronics, especially in high current applications or in applications requiring low resistance levels (e.g., low-noise circuits) [12].

Typical examples of dielectric/insulating materials are glass (inorganic), ceramics (inorganic), plastics (organic) and paper (organic). Insulators are widely used in electronics, and they are crucial elements of widely used components such as thin-film transistors, sensors and interconnections. Low-cost organic dielectric materials that are available in large quantities and can be dissolved in various solvents and solutions can be printed easily as compared to their inorganic counterparts [12].

In printed electronics, polymers, polymer composites, and small molecule-based semiconductors are very important. Semiconductive materials are widely used in transistors, solar cells, light-emitting diodes and sensors. Similar to the conducting materials discussed above for printing technologies, organic/inorganic semiconducting materials are also used for printable sensors and electronics. Inorganic materials have superior properties in terms of performance and stability, while solution-processable organic semiconductors are attractive due to low-cost processing in an ambient environment and flexibility [12].

#### 3.2.2. Substrate Materials 

There are many substrate materials, such as plastics, paper, textiles, glass, and metals, each with its own pros and cons. Most of the time, the substrates for printed electronics need to be cheap and flexible, so papers and polymer films (PET, PEN) are commonly used. The advantages of paper as a material over polymer films are its recyclable nature, its lower cost, and its lower thermal expansion. Papers are limited in their use by their roughness and absorption, but different coatings improve this situation. When it comes to incorporating functionality into clothing, textiles also offer an intriguing alternative. In Roll-2-Roll manufacturing, dimensional stability and mechanical properties of the substrate are also important. In cases where greater thermal and chemical stability is needed, more expensive plastics can be used, such as polyimide, PETS, and PEEK. Glass substrates continue to provide excellent barrier properties, but even flex glass varieties pose some challenges in R2R processing [12].

However, these very thin and flexible substrates tend to have thermal conductivity issues when they are used for applications with high power dissipating profiles. The low thickness and low thermal conductivity of these substrate materials can cause poor heat spreading performance while, factors such as excessive thermal expansion and low glass transition temperature can change the physical and chemical properties of the substrate materials under the heat [13].

### 3.3. Printed Components for Light-Based IoT

Thinking about the flexible and thin realisation of LIoT systems, we will now look at some of the main components of a LIoT node in their printed form—namely, organic photovoltaics (OPVs/solar cells), batteries and light-emitting devices. Printed gas, temperature and pressure, etc., sensors, which undoubtedly are at the heart of both traditional IoT and current trends in printed electronics, will not be considered in detail here, since the topic in this paper is more leaning towards the communication and energy harvesting parts of the system, thus ignoring the specific sensory function of the node.

#### 3.3.1. Solar Cells for Indoor Energy Harvesting 

Billions of IoT devices are expected to be installed over the coming years, with almost half of them connected inside buildings. Currently, the use of batteries to power these devices places significant constraints on this development. Energy harvesting has the potential to solve these hardware issues, providing greater reliability and operational lifetimes in wireless sensor networks. 

Photovoltaic energy harvesting, which is the conversion of light into electrical energy, is the most common method of harvesting energy and is now a well-established technology. The amount of energy harvested depends on the intensity and spectral content of the light falling on the surface of the solar cell, the incident angle of the light, and the size, sensitivity, temperature, and type of solar cells used [14].

PV cell technologies such as Dye sensitised solar cell (DSSC), organic photovoltaics (OPV), copper indium gallium selenide solar cells (CIGS) and perovskite can be manufactured by using printing technologies [15]. According to the solar cell efficiency tables in [16], the currently achieved maximum power conversion efficiency of these PV technologies can be summarised as in Table 1.

Recent work using a new high bandgap organic semiconductor called (IO–4Cl), demonstrates organic solar cells with a PCE of 26.1% measured under the indoor illumination conditions, simulated by a 2700 K LED lamp at 1000 lux [17]. In comparison with the competing photovoltaic technologies, such as traditional c-Si and newly emerging perovskite solar cells, these new OPV devices are the most compatible for indoor applications. In fact, the OPV materials have great tunability of their absorption spectra, thus, it is very easy to match them with the majority of light sources currently used for indoor lighting, which are fluorescent lamps, incandescent lamps, and white light-emitting diode (LED) lamps [18].

#### 3.3.2. Light-Emitting Devices

The most prominent alternatives for light signal production in an all-printed LIoT node (i.e., ignoring solid-state LEDS/micro-LEDs and quantum dots) are the alternating current powder electroluminescent (ACPEL) device, the organic light-emitting diode (OLED) and the light-emitting electrochemical cell (LEC) thanks to thin flexible and printable layers. For example, ACPEL devices can be screen printed establishing economic, simple manufacturing. Their high AC-driving voltage makes them well-suited for energy harvesting applications. However, OLEDs are much more well-suited for low-power LIoT thanks to their low DC-driving voltages. The high sensitivity to oxygen and water of the OLED materials, however, necessitates a high barrier encapsulation. Unlike OLEDs, LECs realise a balanced charge injection using mobile ions in their active layer, broadening the choice in electrodes, but the improvement in lifetime is limited because of incomplete knowledge of the working mechanisms [19].

### 3.4. Printed Electronics for a Sustainable Future

The consequences of the digital transformation and consumer behaviour result in massive amounts of e-waste produced globally. For instance, in 2019, approximately 53.6 Mt of e-waste was generated, and it is increasing at an alarming rate of 2 Mt per year. It is common for e-waste to end up in waste dumps, despite the declared percentages of recycling (20–25%); this has negative impacts on the environment and human health [20].

Organic materials often used in printed electronics are uniquely suited to produce electronics that can be sustainable and biodegradable. Many organic materials have been shown to be biodegradable, safe, and non-toxic, including compounds of natural origin. Additionally, the unique features of such organic materials suggest they will be useful in biofunctional electronics, demonstrating functions that would be inaccessible for traditional inorganic compounds. Such materials may lead to fully biodegradable and even biocompatible and metabolisable electronics for many low-cost applications. For example, numerous materials with a bio-origin have been identified as suitable substrates for the fabrication of organic electronics. Materials such as paper, silk and gelatin enable several functionalities: low-cost, non-toxicity, biodegradability, and often biocompatibility and bioresorbability. Based on these findings, recent research has shown that ‘green’ organic electronics can be very practical and is likely to have a positive impact in the future as it uses biomaterials to make the recycling of e-waste more efficient [21].

For printed electronics to be environmentally friendly, a comprehensive assessment during the device lifetime, known as life cycle analysis (LCA), is required. A life cycle assessment of printed electronics provides a quantitative measure of the environmental impacts at each stage, from production to end-of-life. Researchers have conducted several studies that cover the life cycle stages of printed products and systems. The environmental assessment of printed antennas, as well as assessments of toxic emissions and ageing of polymer films, have been investigated by several research groups. In addition, reports have been published on LCA and recycling options for photovoltaics and the energy demand and reusability of biodegradable and recycled printed LED foils. Similar studies have been conducted for OFETs, carbon-based inks on thermoplastic substrates, and recyclable sensors and TFTs composed of fully carbon-based inks and cellulose-based substrates. While there are limitations, the advancement in printed electronics shows there is a possibility of producing electronic devices with a low carbon footprint [20].

## 4. Requirements and Challenges

In this section, some requirements and challenges for the development of LIoT are identified and discussed. In addition, possible technology alterations according to the design requirements are suggested in the text. 

### 4.1. Eye-Friendly Data Transmission

When using indoor luminaries for illumination and communication purposes, it is essential to maintain eye-friendly, flicker-free and colour consistent indoor illumination while transmitting data. The IEEE 802.15.7 standard for short-range optical communications [22], mainly highlights flicker mitigation and dimming support requirements for short-range optical communications. According to standards, dimming support describes a VLC system that supports changing the perceived brightness of the lights according to the user’s requirement. Moreover, the brightness changing intervals should not exceed the maximum flickering time period (MFTP), which expresses the maximum time period the human eye cannot perceive brightness fluctuations. In [23], MFTP is approximated as a time period within 5 milliseconds. Unregulated flickering VLC transmission could trigger illnesses such as photosensitive epilepsy in sensitive people [24]. The flickering rate of the transmission relies on the modulation scheme used for data transmission. According to [25], modulation schemes such as on-off keying (OOK), colour shift keying (CSK), pulse position modulation (PPM) and orthogonal frequency-division multiplexing (OFDM) are some of the compatible modulation schemes with VLC. However, in [26], authors mention that multicolour mixed white LEDs used for CSK modulation-based VLC systems are more prominent to phenomena such as colour shifting caused by unequal ageing between multicolour sources, which could affect the performance in the long term.

Moreover, in VLC systems, which use RC filter-based pre-equalisers at the transmitter, modulation schemes such as OOK with more power spectral density (PSD) at the DC level are more prominent to cause baseline wandering phenomena (BLW) at the receiver [27]. This could randomly degrade the BER performance of the VLC link based on the transmitting bit sequence. The modulation schemes derived from PPM tend to be immune to BLW as they have less PSD at the DC level. 

### 4.2. Harvesting and Storing Energy at Low Voltage Efficiently

In order to design low powered, energy-autonomous IoT nodes, it is important to select the suitable energy harvesting usage protocol for the application’s requirement. There are three main types of energy harvesting usage protocols, namely Harvest-Use (HU), Harvest-Store-Use (HSU) and Harvest-Use-Store (HUS). As the name suggests, there is no energy store in HU, while there is an energy store step in HSU and HUS [28]. Even though the energy storage-less, simple HU protocol is more supportive of the concept of “expose and connect” based LIoT node designs, energy storage-based protocols have the ability to cater for the power requirement of most currently available microcontrollers and sensors. Dynamically varying power requirements for different functionalities of the node can be smoothly rectified by using energy storage. The structure of HSU can be described as in Figure 2.

For energy-autonomous design, it is vital to obtain the maximum possible energy from harvesters and store them with minimum energy losses and leakages during the operation. Typically, these designs consist of energy harvesting units such as photovoltaics, piezoelectrics or thermoelectric modules, and energy-storing units such as capacitors, supercapacitors, or batteries. When it comes to photovoltaics-based energy harvesting, varying incident illumination on PV cells could result in fluctuations at the storing voltages. Generally, a charge pump circuit or boost converter circuit could increase the final storing voltage at storage in situations such as this. Typically charge pump circuits double or multiply the input voltage by using capacitors, while boost converters use inductors for voltage increase. According to [29,30], charge pump designs are more suitable for steady input, more compact, the on-chip solution required designs, and boost converters are more suitable for high efficiency required designs with fewer space constraints due to the external inductors. Therefore, a boost converter could be the best solution for energy-autonomous designs due to their high efficiency compared to the charge pump approach. Equation (1) expresses the connection between PV terminal voltage and the DC-DC converter voltage output:(1)$$V_{out}=\frac{1}{1-D}\;V_{PV},\;$$
where, *V_out_* = output voltage of the converter, *V_PV_* = voltage at PV terminals, *D* = duty ratio.

Since PV cells are nonlinear, their efficiency can vary due to the level of irradiance and ambient temperature. Therefore, operating them at the maximum power point (MPP) is necessary to achieve maximum efficiency. For this task, the maximum power point tracking (MPPT) method can be used. In this method, the voltage at the terminal of the PV cell is regulated to keep it closer to the MPP corresponding voltage (Vmpp) of the PV cell at the moment. To determine the maximum power point, controllers such as P&O, incremental conductance, fuzzy logic, and artificial neural networks can be used [31,32,33,34,35]. Based on the controller’s output signal, MPP is achieved by regulating the connection between the voltage up-converter and PV cell terminals [36,37]. 

On the other hand, the required surface area for energy harvesting can be minimised by using high-performance PV cells with optimal conditions. The number of PV cells and their series or parallel connectivity configuration can be selected based on the voltage or current requirement of the application. Since the human eye is less sensitive to wavelengths corresponding to the blue colour region of the spectrum, studies in [38] show that blue colour light has to emit more radiant power to achieve a specified lux level, compared to red and green colours. The comparatively more radiant power of the blue colour leads to more energy generation at the PV cells compared to other colours. Therefore, it is beneficial to use luminaires that emit wavelengths of the blue region when designing VLC transmitters for energy harvesting purposes in the indoor environment. According to [39], even though it is an electroluminescence process, Wien law can be used to estimate colour temperature based on the required wavelength region. Using the Wien equation below (2), both eye and energy harvesting friendly suitable light types can be determined.
(2)$$\lambda_{max}=\frac{hc}{4.965kT}$$
where, *h* = Planck’s constant, *c* = velocity of light, *k* = Boltzmann’s constant, *T* = color temperature.

When designing a low-powered IoT node, selection of energy storage type should be made according to the application’s nature. Generally, batteries and supercapacitors are used for these kinds of energy-storing requirements. Among these, batteries are capable of having more energy density, while capacitors have more power density. The high energy density storage devices are more capable of providing steady energy for a long period of time, while high power density storage devices can quickly get energy in or out from the device [40]. Therefore, to power up the “expose and connect” concept based LIoT, it is important to minimise the storage device’s initial cold start charge time and charge quickly to starting voltage. By using high power density storage such as supercapacitors, this requirement can be fulfilled. The voltage response for charging a supercapacitor can be expressed as the below (3).
(3)$$V_{c}\left(t\right)=V_{in}\left(1-e^{-\frac{t}{\tau }}\right),$$
where, *τ* = time constant defined as
*Τ* = *R* ∗ *C*,(4)
where, *C* = capacity, *R* = series resistance.
(5)$$C=\frac{Q}{V}\;,$$
where, *C* = capacity, *V* = voltage.

According to (3), when time is equal to one time constant, the capacitor is charged up to approximately 63% of its final voltage. Similarly, the capacitor gets discharged to 37% of the final charged voltage during a one-time constant equal period. On the other hand, the equation explains that capacity value is directly proportional to the accumulated charges at a given voltage. Therefore, it is essential to keep the optimum resistance and capacity values when selecting a capacitor [41]. The equal series resistance (ESR) of the capacitor can be identified by the ESR rating value. By using a low ESR rated capacitor with a sufficient capacity rating for the required application, the time constant can be minimised.

In addition, according to (5), it is important to keep sufficient voltage at the storage unit to store more charges. However, when supplying the energy to a connected load such as a microcontroller or sensor, the voltage level needs to be reduced. Typically for this purpose, a low dropout regulator (LDO) or buck converter can be used. According to [42], buck converters have more efficiency than LDOs when the load draws milliamperes rated current. However, LDOs are more suitable for compact in-chip designs as buck converters require space-consuming external inductors for their operation. Therefore, a suitable converter needs to be chosen based on the requirement of the application. For applications that involve low power-autonomous designs, using high-efficiency buck converter designs would be beneficial.

### 4.3. Low Energy Consumption Optimisation

In order to preserve the harvested energy while the IoT node is not actively operating, a sleep state can be introduced to the low power node design. In this way, the node is put into a low power consuming intermediate status, which can be woken up with external or internal interrupts. Generally, the duty cycling method can be effectively used for this purpose. In this method, nodes periodically alternate between the sleep and wake up modes. In addition, duty cycling can be mainly classified as synchronous and asynchronous approaches. In duty cycling, nodes self-wake up periodically so that the transmitting device has to wait till the scheduled node wakes up time to communicate. However, the transmission delay between the states’ alternative period is considered the main drawback of this method. On the other hand, the wake-up radio (WuR) approach can be used for on-demand wake-up purposes without causing any delay. Typically, this approach uses a low power consuming simple modulated secondary receiver unit for the node’s sleep interruption. In [43], the authors describe an energy-autonomous VLC based WuR receiver with a pattern correlator. In this way, the WuR receiver can filter out the dedicated wake-up signal, among many other wake-up signals. Since a secondary receiver unit is active throughout the sleep state, this method has less power efficiency compared to duty cycling [44]. Therefore, considering the trade-off between transmission delay and less power efficiency, these methods need to be integrated with the node designs. 

### 4.4. Early-Stage of Sustainable Electronic Technologies

In contrast to traditional electronics, sustainable electronics technologies are still early in their development process. Hence, finding specific active and passive components for developing an entire prototype using printed or biodegradable electronics appears to be a true challenge at this time. However, according to [45,46,47], essential components for LIoT nodes, such as printed photodiodes, OLEDs, indoor PV cells, displays, and temperature sensors, are currently available as stable components for the integrations. Therefore, it is clear that an early-stage LIoT prototype development process can be initiated using traditional and sustainable electronics-based components. With the development of the technologies in future, these traditional electronics can be replaced with their alternative counterparts, and a fully printed LIoT node will be implemented.

## 5. Design of a Duplex LIoT Node

In order to demonstrate the concept of LIoT, we have designed and implemented a working prototype. The design is a LIoT-based smart product label, which contains display and temperature monitoring sensors. When the node is exposed to the transmitter’s illumination, the product label is expected to update its information display while transmitting the temperature variation of the wrapped product. The design of the node can be illustrated as in Figure 3.

The smart label was designed to work as a duplex communication system with both receiving (i.e., downlink) and transmitting (i.e., uplink) capabilities. In addition, the system is designed to support multiple users using the same transmitter illumination infrastructure (multi-user operation). In this prototype, visible light and infrared links are used as the downlink and the uplink, respectively. During the idle period of time, the node is configured to go to the energy-saving sleep state, and whenever the downlink is operating, the node is designed to wake up and receive data.

### 5.1. Transceiver—Illumination System

The overall structure of the transceiver-illumination system is illustrated in Figure 4.

The input data from the main computer is initially transferred to the microcontroller unit (MCU) by using serial communication. From the MCU, the data is encoded and modulated according to the selected modulation. After that, the modulated data is converted to an electrical signal, and it is fed to the LED driver to control the transmitting lights according to the signal. Inside the LED driver, the electrical signal is levelled up to provide sufficient voltage and current level required by the selected lights. On the other hand, the transmitted optical signals from nodes are received from the receiver at the transceivers surface. The generated electrical signals based on the received signals are transferred to the MCU for decode and demodulation.

In order to select the modulation scheme of the prototype, factors such as VLC compatibility and robustness against capacitive component-specific phenomena such as BLW are considered. In the implemented prototype, pulse position modulation (PPM) was used as the digital modulation scheme for the optical transmitting signal. VLC friendliness and the modulated data frame is designed to facilitate 48 bits, containing address bits and payload bits for each targeted LIoT node. Moreover, additional on-off keying (OOK) based wake-up bits are added in front of the data frame to wake up the node from sleeping mode. In addition, to minimise the interferences from external sources, the transmitting signal was modulated using a 38 kHz carrier signal. In the final design, an Atmega328P based Arduino Uno development board was used as the MCU of the transceiver prototype. A TSOP 1838 IR sensor is used as the IR receiver of the transceiver. A captured waveform of a data frame is illustrated in Figure 5.

The LED driver of the transceiver was designed to provide the required DC bias voltage for lights while detecting the MCU’s TTL level voltage signals and scale them to create better modulation depth. Moreover, the required DC bias voltage, and factors such as rise fall time, drain current limitation, and compatibility with TTL level signal was considered for the design of the LED driver. Based on the above factors, an IRF520 N channel metal–oxide–semiconductor field-effect transistor (MOSFET) was used as the switching unit for the LED driver. The design diagram of the LED driver is illustrated in Figure 6.

As the light-emitting sources, indoor illumination and energy harvesting friendly 5700 K—cool white chip on board (COB) type LED sources were used. In order to mitigate the perceivable flickering effects and enhance the energy harvesting performance, an idle transmitting mode was introduced to the system. In previous work, we have shown that idle transmit mode can improve the energy harvesting mode compared to relying on bias voltage [48]. In the idle transmitting mode, when the transmitter is not transmitting any data signals, unmodulated long square pulses are fed to the LED driver and transmitted through the lights. In this way, voltage fluctuation on PV cells can be minimised between the data transmitting and non-transmitting interval.

### 5.2. LIoT Node

The structure of the prototype node is depicted in Figure 7. The node consists of main segments such as transceiver system, energy harvesting system, temperature sensor, MCU and output printed display.

The designed algorithm for the LIoT smart label is described in Figure 8. Initially, the node receives the optical signal and checks the request correspondence for the node’s main functions. Sensing and transmitting the temperature and managing the info display are the two main functions of the designed prototype.

#### 5.2.1. Transceiver System

The transceiver system consists of a visible light receiver and infrared emitter to facilitate downlink and uplink functions. In this design, an on-demand wake up capable WuR inspired wake-up approach was used for energy conservation. Since the VLC receiving sensors have better power efficiencies compared to RF receiving antennas, the same receiving sensor is used for wake-up signal reception in the implementation. This has mitigated the need for a separate receiver when using the WuR approach, thus, improving the overall power efficiency, the common deficiency of the approach. As WuR approach already has less transmission delay compared to its duty cycle approach-based counterpart, this has resulted in a solution that contained the best of both approaches. The visible light receiver was designed to receive the downlink data signal and extract the 38 kHz modulated signal in order to demodulate and decode. In this circuit, initially, the photodiode receives the optical signal and creates a corresponding current signal. After that, a trans-impedance amplifier (TIA) converts the current signal to a voltage–electrical signal. Then the electrical signal is amplified by using an automatic gain control amplifier (AGC). By analysing the header, the AGC adjusts the gain level and provides the waveform for the bandpass filter with 38 kHz centre frequency. In this way, the bandpass filter filters out the external noise components from the converted electrical signal and passes only the message electrical signal for processing. During AGC processing, the electrical signal is modified to TTL voltage levels to be detected at MCU’s analogue to digital converter. The MCU algorithm initially waits for the wake-up interruption and then identifies the data signal measuring the header time length. Thereafter, MCU is configured to demodulate and decode the message signal and proceed according to it. For the implementation, a 3.3 V voltage compatible ATmega328P based Arduino mini development board was used for the MCU. TIA, AGC and bandpass filter all on-chip integrated VSOP 38338 integrated chip (IC) with 525 nm PIN diode used for the receiver prototype. The structure of the receiving system is given in Figure 9.

In order to minimise the uplink-downlink optical channel interferences and avoid naked eye perceivable flicker, which could be caused by a visible light emitter, an IR emitter was used as the uplink for the node. The uplink transmitting unit was designed to transmit temperature sensor readings using infrared (IR) light. According to the design, the voltage variations provided by the 3.3 v compatible TMP36GT9Z temperature sensor are fed into the MCU for temperature mapping. Thereafter, the processed temperature information is encoded and modulated for uplink transmission. Similar to the downlink, uplink data are also modulated using a 36 kHz carrier signal in order to mitigate the noise effects. As the information display of the smart label prototype, a flexible printed electrochromic seven segment display was used. The bistable nature of the electrochromic display helps to hold the display information during node sleep state, which is advantageous for a low power product label design. Since the display is 3.3 v compatible, it was directly connected to the digital output pins of the MCU unit.

#### 5.2.2. Energy Harvesting Unit

In order to achieve energy-autonomous, battery-less design, the energy harvest unit (EHU) of the prototype is designed to keep minimum energy storage on the device while providing a continuous power supply for the node to operate. The overall design of the EHU followed the type of HSU energy harvesting and usage protocol to facilitate the fluctuating power requirement of the commercial MCU and sensor nodes. The designed EHU consists of two PV cells, a power management IC (PMIC) with a built-in boost converter, prismatic capacitor and buck converter to support the energy harvest process. The structure of the designed EHU is illustrated in Figure 10.

In the EHU, two parallelly connected Epshine LEH printed organic PV 8 cell modules were used to harvest sufficient energy for the requirement. Then, with the use of a boost converter and the MPPT tracking feature integrated with the e-peas AEM 10941 PMIC, the input voltage further improved while keeping PV cells at the maximum efficiency region. A low ESR 0.4F supercapacitor manufactured CAP -XX was used as the harvested energy storage of the node. According to the configured algorithm, whenever storing voltage reaches 3.9 v, PMIC turns on the buck converter to step down the storing voltage to 3.3 v voltage. For this, a Texas TPS 62740 step-down buck DCDC converter was used for the implementation. When the storing voltage reaches 3.3 v at the capacitor, the PMIC turns off the buck converter and detaches the load to speed up the charging process. The designed energy harvesting system’s turn on-off capacitor voltages can be tabulated as in Table 2.

During this turn on-off period of the window, the node can be fully functional and perform its designed task. The state transition and its dependable parameters are illustrated in Figure 11a. The final view of the designed node is shown in Figure 11b.

## 6. Performance Evaluation

In this section, key performance experiments and results are described. The main functions of the LIoT concept are experimented with and evaluated here. Based on the experiment results, some hardware components selection were made for the prototype implementation. Since the prototype of LIoT smart label does not require a higher data rate for its functionality, approximately 1.4 kbps (max) of low data rate was used for the communication. 

### 6.1. Uplink—IR LED (5 mm) vs. Printed OLED

In order to compare the performance of a traditional IR uplink (Lambertian radiation pattern) with VLC and printed OLED emitter-based uplink (non—Lambertian radiation pattern) under the LIoT smart label usage scenarios (on market shelves, inside refrigerators) the following experiment was carried out. The performances of the two types of links under aligned and non-aligend transmitter-receiver alignment scenarios were tested during the experiment. To simulate a possible LIoT label-based product displacement during an actual application scenario, a 15 cm horizontal displacement was introduced. For the implementation, a 5 mm IR LED and 4 cm × 7 cm printed white OLED was used as the transmitting sources. A monolithic photodiode and on-chip trans-impedance amplifier integrated OPT 101 analogue sensors were used as the receiver for both setups. The setup arrangement is depicted in Figure 12a. In this experiment, the IR and printed OLED transmitters were initially placed directly facing the photodiode sensor, and the SNR was measured. The transmitters were placed 0.20 m away from the receiver sensor, and minimum external illumination conditions (below 20 lux) were maintained during the experiment. Then, the receiver sensor was moved 15 cm horizontally away from the previous position making the transmitter and receiver nonaligned, and the experiment was repeated. The results are plotted in Figure 12b.

The results suggest that a typical 5 mm IR LED based uplink has a better frequency response than the printed OLED when both transmitters and receiver sensors are placed directly. However, during the non-direct experiment, the results showed that the signal emitted by the planar lighting source (printed OLED and sensor act as non-directional link) has a better frequency response than directional lighting source (5 mm IR led and sensor act as a hybrid link) when they are not spatially aligned. The Lambertian radiation pattern emitted by the 5 mm IR led causes more concentrated radiation on the receiver in the aligned link scenario, while the non-directional, wider emitting angles of the OLED results in better performance when links were not aligned. 

Therefore, according to the results, printed OLED with higher emitting angles would be a good choice for more dynamic, close-ranged LIoT applications. The large irradiation pattern caused by the high emitting pattern could provide stability compared to the more concentrated radiation pattern. On the other hand, the concentrated directional IR link can be advantageous for stationary applications such as smart labels which are considered in the implementation. The high SNR levels of the IR link could result in more robust communication links under noisy environments. 

### 6.2. Frequency Response of Printed PV Cells

In order to determine the feasibility of using commercially available printed PV cells for LIoT data receiving purposes, the frequency responses of different printed PV cell types were measured in this the experiment. To create a similar environment as in the previously discussed experiment, the PV cells were placed 0.2 m below the transmitter, and square pulses were transmitted as the signal. Indoor optimised Perovskite PV cells and both indoor and outdoor optimised organic printed PV cells were used in this measurement setup. Since each printed PV cell panel had a different active area, the normalised voltage per square centimetre of the area was considered for the SNR calculation. The details of used PV cells are given in Table 3. The obtained results are plotted in Figure 13.

According to the results, both perovskite and organic printed PV cells show low SNR performance when the frequency increases. Therefore, commercially available printed PV technologies such as OPV and perovskite have low-frequency characteristics which are not capable of supporting high data rate applications for LIoT. The designed receiver circuit for the prototype node was not able to demodulate and decode the extracted data signals from the PV cells. Hence, a photodiode based receiving circuit was added to the prototype for high data rate receiving purposes.

### 6.3. Effect of MPPT and Boost Converter on Indoor Energy Harvesting

The feasibility of using dynamic outdoor sunlight energy harvest optimising technologies for the static transmitter based LIoT application scenarios were tested in this experiment. In order to simulate the receiving levels with dynamic movements of LIoTs under the fixed transmitter, the illumination of the transmitter was varied with 120 s fixed interval during the experiment. The MPPT and boost converter’s effect on the LIoT node’s energy harvesting process was analysed during this experiment. The MPPT-based and boost converter-equipped AEM 10941 PMIC and non-actively energy harvesting contributing LTC3588 PMIC, were used for this experiment. To store the harvested energy, a cap-xx high-performance 0.4 F supercapacitor was used. In the experiment, the illumination levels were changed with a fixed interval of 120 s, and the voltage between the capacitor was measured. The experiment was repeated for both types of PMICs. The obtained results are plotted in Figure 14.

The results suggest that under low lux levels, MPPT and boost converter managed to store more energy compared to other approaches while keeping PV panels at optimal efficiency. However, with the increasing illuminance levels, the gradients of the graphs show that the advantage of MPPT is negligible compared to low illuminance levels. This can be explained by the fact that higher illuminance on the PV cell naturally drives the cell closer to the MPP so that the effect of the algorithm based MPPT controller is less dominant under these conditions. 

Therefore, using MPPT and boost converter setups will be more viable for LIoT applications designed to operate under dynamic light-receiving conditions. In addition, since MPPT and boost converter equipped PMICs have a limited input voltage range (typically 0–5 v), extra precautions need to be taken to avoid overvoltage generating scenarios. For the static LIoT applications, fixing the ideal illumination levels at the designing stage could achieve maximum energy harvesting at the PV cells. In this way, static LIoT applications can avoid additional PV optimising circuitries and follow a less complex energy harvesting design approach. 

### 6.4. Performance Analysis of the Designed Energy Harvesting Unit

In order to assess the performance of the designed energy harvesting system, the following experiment was carried out. A 5 mm IR LED was used as an uplink emitter for calculating the transmitting current requirement. The max ratings of the drawing currents for each operating mode are listed in Table 4.

In order to create a fairly high current consumption for the evaluation, the node was programmed to operate in transmitting and receiving mode in equal operating periods without falling to sleep mode. Initially, while the node is completely turned off (zero consumption), the charging time from Vcp 3.3 v to 3.9 v was measured (described in Table 2). Even though the PMIC has the ability to charge the Vcp up to 4.5 v under no-load conditions, to determine the minimum operational time values, a turn-on value of 3.9 v is considered as the maximum charging Vcp throughout the experiment. Then, as soon as the node starts to turn on when Vcp reaches to 3.9 v, the discharging time from 3.9 v to 3.3 v under the given illuminance was measured. The experiment was repeated by varying the illuminance on the two parallelly connected printed organic PV cells during the experiment. The received results are plotted in Figure 15a.

According to the results, the node’s energy harvesting system works at maximum performance when the illuminance levels are relatively higher. Even though the energy harvesting and storing system manages to provide only around 30 s of minimum operational time for 800 to 1200 lux illuminance levels, it is fully sufficient for the functionality of the LIoT smart label application. As previously mentioned, by keeping the node at a sleeping state (no load) throughout the charging time, it can achieve a maximum capacitor voltage of 4.5 v. Hence, the operational time can be further increased without any additional effort. 

By using the values obtained from the previous experiment, the operations-to-charge time ratio curve was determined. The equation used to determine the operation-to-charge time ratios given below.
(6)$$operation-to-charge\;time\;ratio=\frac{Minimum\;operational\;time\;per\;given\;illumination\;level}{Charging\;time\;for\;Vcp\;to\;\mathrm{off}\;\mathrm{to}\;\mathrm{on}state\;voltage\;threshold}\;,$$

The obtained curve for the prototype node is illustrated in Figure 15b. 

Using these curves, the node will be self-aware about its limitations under the given illumination condition and can accordingly decide its own operation and sleep times by coordinating with the transmitter. In this way, LIoT nodes can be designed to operate as a “smart LIoT node” which is aware of its current (situational) limitations and has the ability to adapt accordingly to execute required tasks. The proposed algorithm for smart LIoT is illustrated in Figure 16. 

## 7. Discussion

According to the study of Lambertian vs. non-Lambertian optical emitters usability for LIoT, the results indicate that lighting sources with small emitting angles can achieve better SNR rates for stationary LIoT applications, while sources with large emitting angles are more suitable for dynamic LIoT applications, where mobility is expected. The frequency response comparison of two organic and Perovskite printed PV technologies showed that printed PVs have a slower frequency response for LIoT signal receiving purposes. Furthermore, the experiment on the applicability of MPPT and DC-DC boost converter-based energy harvesting for LIoT highlighted that MPPT and boost converter could improve the energy harvesting efficiency in low light conditions. The operation-to-charge time curve derived from the EHU’s performance analysis results can be used to implement a novel limitation awareness smart algorithm for the LIoT. In this way, the smart LIoTs will execute tasks adoptively under the varying power constraint scenarios. 

## 8. Conclusions

A feasibility study and implementation work of the energy-autonomous LIoT node has been presented in this paper. In this work, a LIoT based smart label prototype was implemented as a case study. Based on the performance and results of the prototype, the practicality of the proposed LIoT framework was analysed. The use of visible light for the LIoT nodes’ communication enables indoor energy harvesting through the same infrastructure while adding unique VLC advantages over radio communications. In order to support the idea of the sustainable green communication concept, the usability of sustainable implementation technologies for the implementation was considered throughout the work. The printed technologies-based components used for the implementation, such as display, OLED and PV cells, worked as expected with other conventional components and sensors, despite their lower performance. Technologies such as visible light communication and self-energy harvesting are getting attention to cater to the requirement of sustainable zero energy nodes, while massive wireless energy transfer is expected to be a key technology in the new 6G IoT. As for the future development of this work, inter-nodes power-sharing approach will be developed and implemented. In addition, the smart LIoT concept can be further improved considering adaptive transmitter illumination level adjustments and internode transmission optimisation with proximity information. With the development of thin electronics technologies integrable on any surface, such as PE, it will be possible to integrate different functionalities on any available surface. These integrated nodes, sensors, actuators and others, will unlock new boundaryless “living surfaces” technology that does not require any hands-on devices in the future.

## Figures and Tables

**Figure 1 sensors-21-08024-f001:**
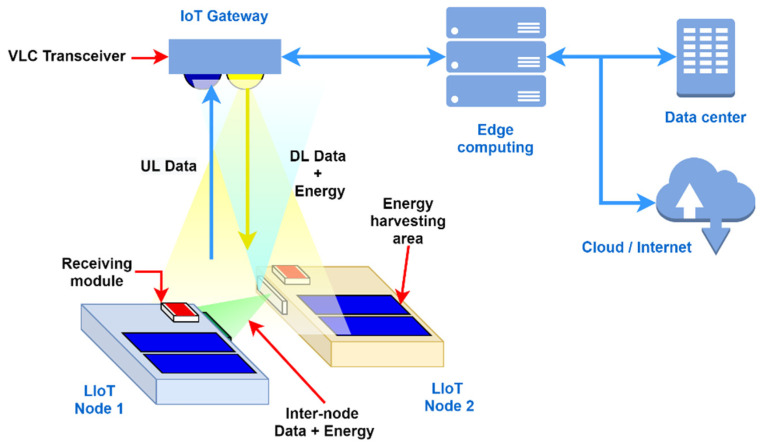
The light-based IoT concept and architecture: example of two networked LIoT nodes.

**Figure 2 sensors-21-08024-f002:**
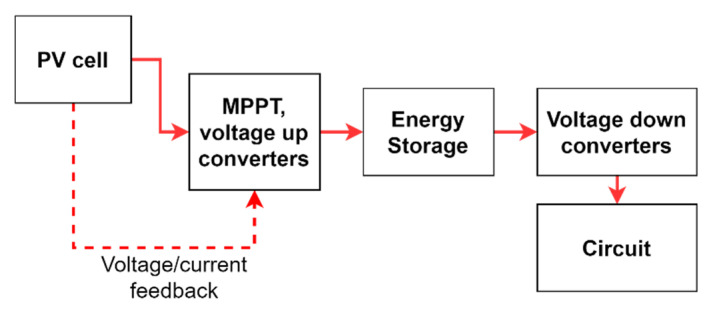
Harvest-Store-Use energy harvesting structure.

**Figure 3 sensors-21-08024-f003:**
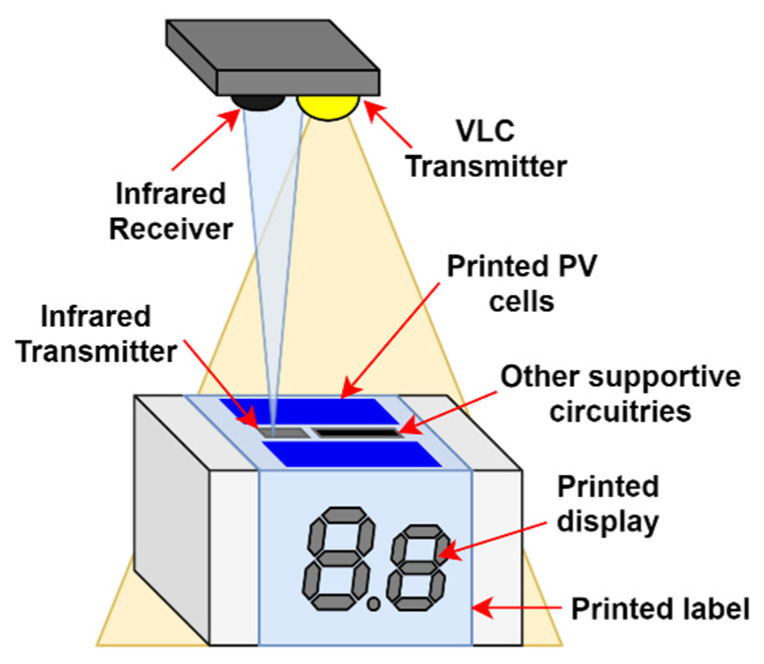
LIoT-based product smart label design.

**Figure 4 sensors-21-08024-f004:**
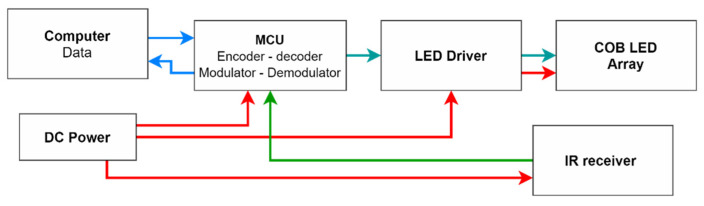
Structure of the transceiver-illumination system.

**Figure 5 sensors-21-08024-f005:**
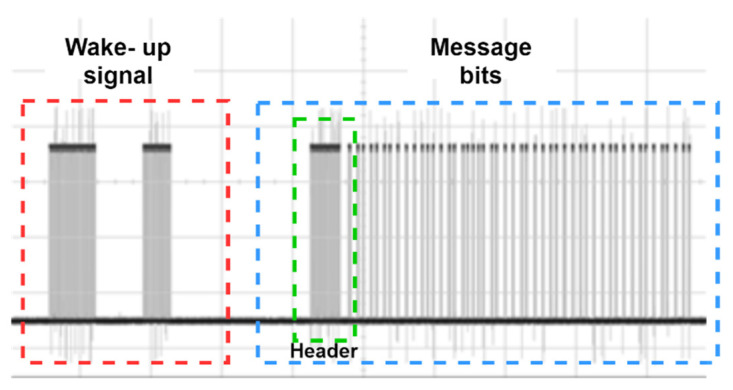
Oscilloscope capture of a designed data frame.

**Figure 6 sensors-21-08024-f006:**
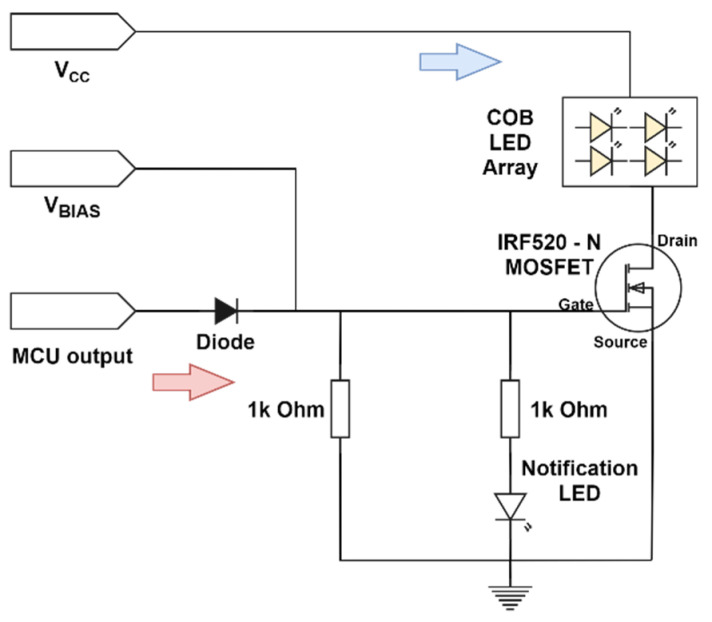
LED driver structure.

**Figure 7 sensors-21-08024-f007:**
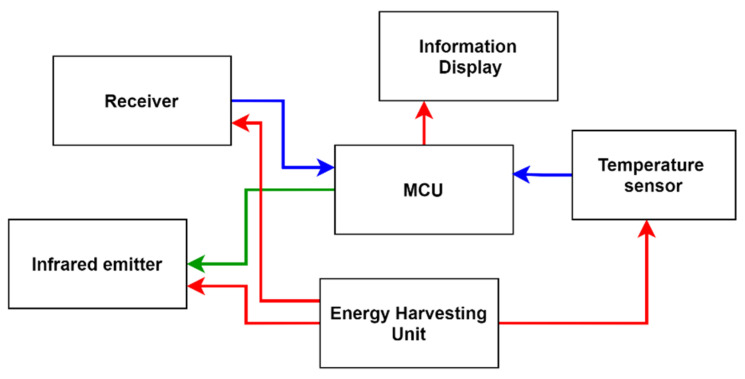
The structure of the prototype LIoT node.

**Figure 8 sensors-21-08024-f008:**
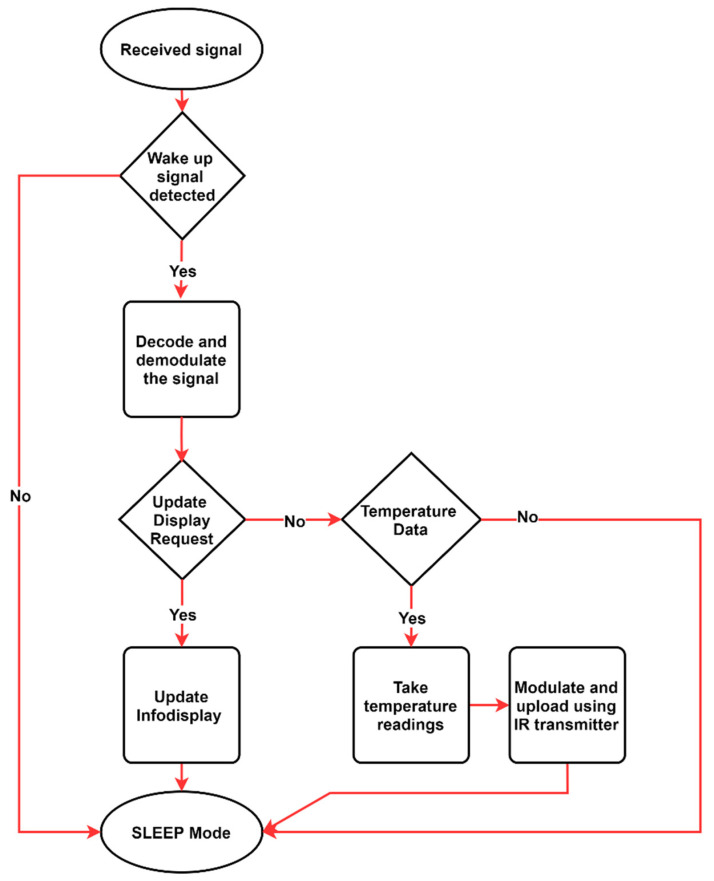
Designed algorithm for the LIoT smart label.

**Figure 9 sensors-21-08024-f009:**
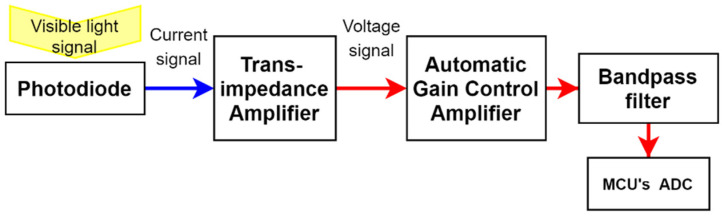
The structure of the receiving system.

**Figure 10 sensors-21-08024-f010:**
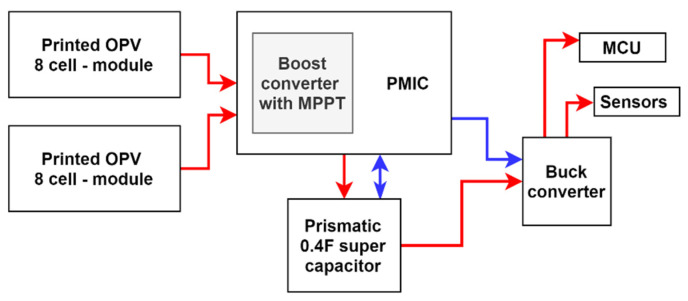
The structure of the designed EHU.

**Figure 11 sensors-21-08024-f011:**
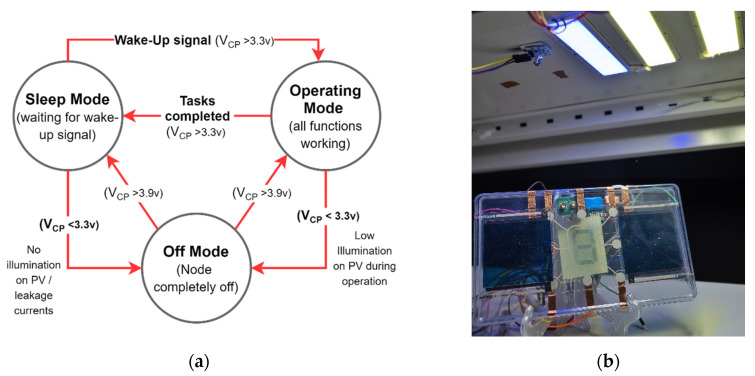
(**a**) State transition diagram and conditional parameters; (**b**) LIoT product label prototype using printed components.

**Figure 12 sensors-21-08024-f012:**
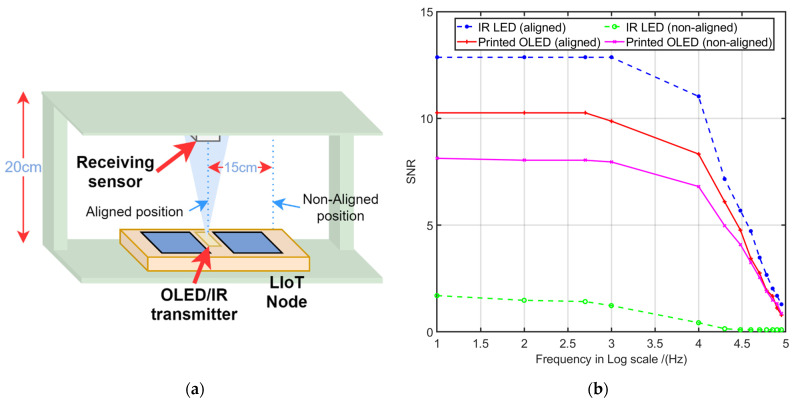
(**a**) Experiment setup arrangement; (**b**) IR vs. printed OLED under different link alignments.

**Figure 13 sensors-21-08024-f013:**
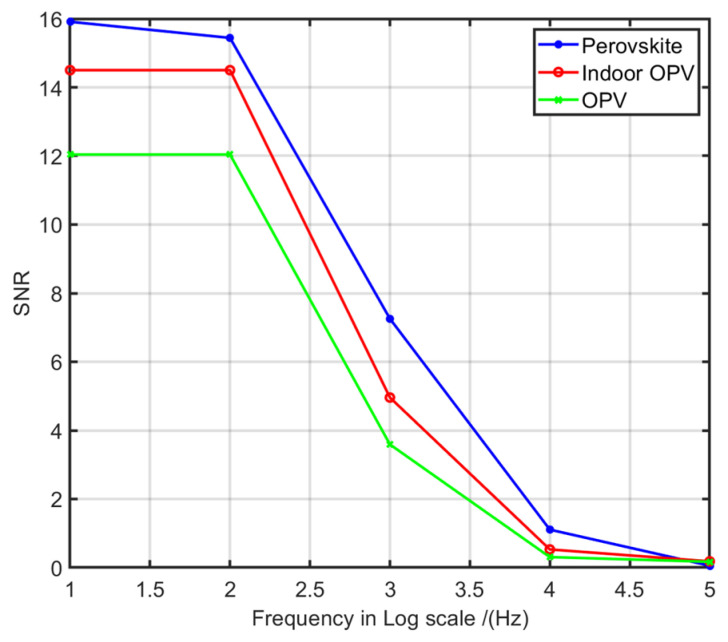
Frequency response of the PE based perovskite and OPV technologies.

**Figure 14 sensors-21-08024-f014:**
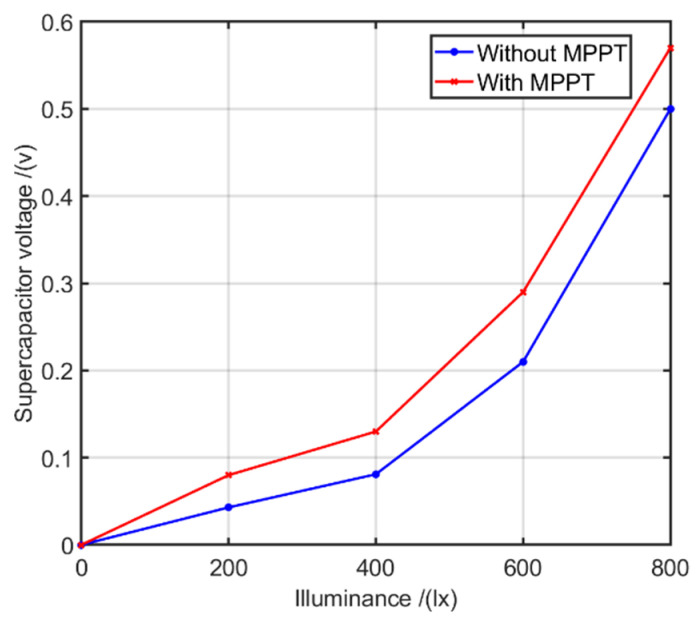
Capacitor voltage variation with and without MPPT.

**Figure 15 sensors-21-08024-f015:**
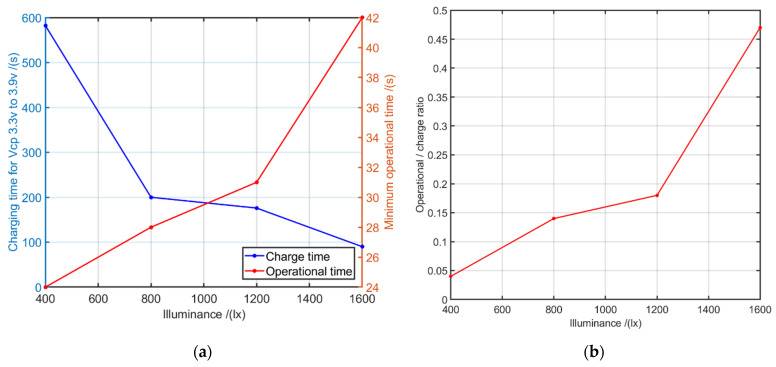
(**a**) Charge time and operational time variation for different illuminance; (**b**) illumination vs. operational-charge ratio variation.

**Figure 16 sensors-21-08024-f016:**
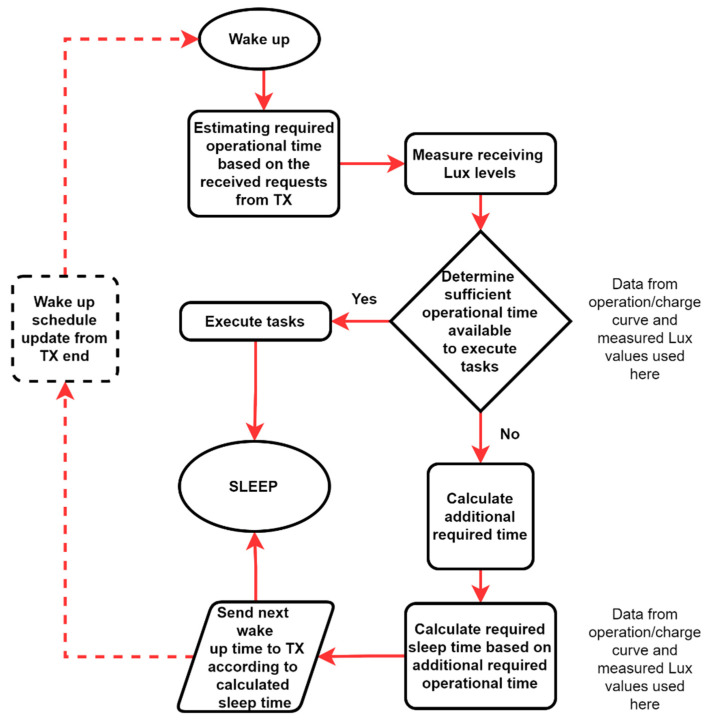
Proposed algorithm for Smart LIoT.

**Table 1 sensors-21-08024-t001:** Power conversion efficiency under global AM1.5 spectrum (1000 W/m^2^) at 25 °C.

PV Type	Power Conversion Efficiency (PCE)
CIGS (cell) (Cd-free)	23.35 ± 0.5
Perovskite (cell)	22.6 ± 0.6
DSSC (cell)	11.9 ± 0.4
OPV	15.2 ± 0.2

**Table 2 sensors-21-08024-t002:** Turn on-off voltages of the designed EHU.

State	Capacitor Voltage (Vcp)
Turn off	3.3 v
Turn on	3.9 v

**Table 3 sensors-21-08024-t003:** Commercially available printed PV cells used for the experiment.

Printed PV Type	Manufacturer	Active Area (cm^2^)
OPV (indoor)	Epshine	35
OPV (outdoor)	Infinity PV	300
Perovskite	VTT	95

**Table 4 sensors-21-08024-t004:** Maximum ratings for different operating modes of the node.

Operating Mode	Maximum Drawing Current
Sleep	<1 mA
Receiving/Displaying	3 mA
Transmitting	25 mA

## Data Availability

Not applicable.

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
