# Peer review of "Light-Based IoT: Developing a Full-Duplex Energy Autonomous IoT Node Using Printed Electronics Technology"

_sensors, 2021, doi:10.3390/s21238024_

Round 1
Reviewer 1 Report
In this paper, authors have studied the “"Light-based IoT: Developing a Full-duplex IoT Node Using Printed Electronics Technology”. It is an interesting topic. However, author should address the following comments before final acceptance. The detail comments are given below:
- Author should briefly explain the IoT system architecture
- Author should also add the commercial aspect of Light-based IoT
- Printed or flrxible diplays have low a low stability. They are mostly based on conductive polymer, and polymeric materials have low thermal and UV-Stability. Please discuss the stability issue in detail.
- The author should specify the type of photovoltaic cell. Because silicon solar cells cannot work in the diffuse or dim sunlight. However, Dye-sensitized solar cells (DSSCs) possess an efficient power output in the entire range of lighting conditions, including LED lighting or indoor fluorescent. Even, they can work efficiently in the conditions of diffused or dim sunlight.
Reviewer 2 Report
The manuscript need be improved a lot before its acception:
- the maintext need be concise, now it is like an experimental report;
- the title is too wide to presenting exactly what the authors did in this work;
- the quality of the figures need be thoroughly improved;
- More importantly, what is the novelty of this work? In the section of introduction, the authors need introduced other previous works on self-powered sensing nodes, which are numours, and make a comparison with these previous work, showing their novelty of the work.
Reviewer 3 Report
Dear Authors,
It is good paper with experimental works and some results are presented. But there is a lack of clarity on the results and also the paper is hard to follow as a reader. There is a lack of connection and follow in the paper.
- Half of the paper (until page ) is background information but it is still hard to follow. Here are some comments to address so that the quality of the paper can be improved.
- Advised to reduce the background to not more than five (5) pages- keeping only essential background information. Some of the content is unnecessary.
- Experimental setup Figure 12 a) - there is only 20cm between Tx and Rx and moving 15cm more in horizontal. It is not clear where are the possible applications in IoT. It is not clear. You need to make connection this experimental setting with applications discussed in the background.
- There is lack of clear explanation of experimental setting. In Figure 12 b), what is the results shown is not clear and need to have clear explanation. How the measurement is done and how the result obtained. You need to include some details.
- Similar comments are applied in the rest of the whole performance evaluation section. The results are not clear and also not linked with the rest of the paper or background information.
Regards,
Round 2
Reviewer 3 Report
Dear Authors,
Well done.
Regards,
Rajan
Author Response
Dear reviewer , Thank you very much.